# Medical Dispute Cases Caused by Errors in Clinical Reasoning: An Investigation and Analysis

**DOI:** 10.3390/healthcare10112224

**Published:** 2022-11-07

**Authors:** Ching-Yi Lee, Hung-Yi Lai, Ching-Hsin Lee, Mi-Mi Chen

**Affiliations:** 1Department of Neurosurgery, Chang Gung Memorial Hospital at Linkou, Taoyuan 333423, Taiwan; 2College of Medicine, Chang Gung University, Taoyuan 333323, Taiwan; 3Department of Radiation Oncology, Proton and Radiation Therapy Center, Chang Gung Memorial Hospital at Linkou, Taoyuan 333423, Taiwan

**Keywords:** medical disputes, clinical reasoning, diagnostic error, medico-legal case

## Abstract

Studies that examine medical dispute cases (MDC) due to clinical reasoning (CR) are scarce in Taiwan. A retrospective analysis was undertaken to review MDC filed at four hospitals in Taiwan between 2011 and 2015. Cases were examined for the healthcare professionals involved, their relevance to CR errors, clinical specialties, and seniority. Seventy-eight MDC were identified and 57.7% of which were determined to be related to CR errors (*n* = 45). Among the 45 cases associated with CR errors, 82.2% (37) and 22.2% (10) were knowledge- and skill-related errors, respectively. The healthcare professionals with the most MDC were obstetrician-gynecologists (10/90, 11.1%), surgeons (8/90, 8.9%), and emergency physicians (7/90, 7.8%). The seniority of less than 5 years or lower had the highest number of attending physicians to be associated with MDC. In contrast, the highest seniority (>25 years) in the physician group and year 6 in the resident group are both shown with zero MDC. In our study, the larger hospitals had a significantly higher incidence of MDC compared to the smaller hospitals (Pearson Correlation Coefficient = 0.984, *p* = 0.016). An examination of MDC reveals the frequency and nature of medical errors in Taiwanese hospitals. Having identified that CR errors contributed a substantial fraction to the overall MDC, strategies to promote reasoning skills and hence reduce medico-legal issues help safeguard both patients and healthcare professionals.

## 1. Introduction

As society progresses toward the internet era, the growing public awareness of patient rights has inevitably led to the likelihood of medical disputes in clinical practice [1,2,3]. In recent years, many countries, including Taiwan, have experienced a surge in medical dispute cases [4,5,6,7,8]. Medical disputes commonly arise from medical errors/negligence, malpractice, adverse events, and miscommunication [9,10]. In the discussion of causation in medical errors, some errors are related to individual factors, instrument malfunctions, or system design flaws [11]. Disputes regarding medical errors may have detrimental effects on patient safety and the doctor-patient relationship. It is important to recognise that some errors are preventable despite the various types and causes of medical errors. Recent studies estimated that preventable medical errors account for over 250,000 deaths in the U.S., resulting in annual national costs of over one trillion dollars [12,13]. The costly consequences and adverse impact of medical errors have received substantial attention globally, calling for rigorous strategies for improving healthcare quality and addressing dispute resolution through the prevention of medical errors [11,14,15].

To err is human [16], and normal human behaviour places every health professional at risk of unexpected medical errors generated from misdiagnosis or a minor mistake. Following the identification of human and system imperfections as the sources of medical errors, research suggests that human diagnostic errors occurring in 10–15% of cases contribute significantly to total medical errors [17,18]. Clinical reasoning is an integral process performed by health professionals to collect and interpret information, which then guides them to think through clinical problems, synthesise an accurate solution, and eventually establish an intervention or management pertinent to the patient’s condition [19,20,21]. Faulty clinical reasoning is considered one attribution to diagnostic errors, and several studies have indicated that insufficient knowledge or skills may play a role in causing reasoning failures [21,22,23,24]. While it is impossible to eradicate human error, attempts at error measurement, such as the analysis of medical dispute cases, have been made to deal with medical uncertainty [25]. A better understanding of the source of these errors will help develop effective approaches to reducing them.

Currently, research on medical disputes caused by errors in clinical reasoning is very limited in Taiwan. This study is intended to perform a retrospective examination of medical dispute cases registered from 2011 to 2015 at the Administrative Center for Legal Affairs Department, located in a local hospital in Taiwan. In this analysis, we aimed to investigate the incidence of medical dispute cases and explore for any correlations to be found between the cases and different clinical departments or specific healthcare professionals. The study also determines how medical dispute cases are related to clinical reasoning and hence provides guidance for policy development or training in improving clinical reasoning skills in order to reduce medical errors or disputes and hence enhance patient safety.

## 2. Method

A retrospective analysis was undertaken to review all the medical dispute cases filed at the Legal Affairs Department located in a local hospital owned by Chang Gung Medical Foundation in Taiwan. This internal department is responsible for the representation of four branches of a private hospital network operated by the foundation in all legal matters. One major branch in northern Taiwan, where the records of hospital litigation were kept and used in our study, is a large teaching hospital and tertiary medical centre with a 4176-bed capacity. The medical dispute cases from the remaining three hospital branches in the northern and southern regions of Taiwan were also documented in this department. Another northern branch is a district hospital with a 1089-bed capacity. One of the southern branches is a district hospital with a 1384-bed capacity, while the other southern branch is a teaching hospital and medical centre with a 2680-bed capacity. Cases that occurred from 2011 to 2015 were examined for the types of healthcare professionals involved (physicians, nurses, and other medical staff), their relevance to clinical reasoning errors, clinical specialty, and seniority. A Pearson correlation coefficient analysis was used to assess the association between the number of dispute cases and hospital size (measured by the number of beds). One physician with expertise in neurosurgery, one qualified legal counsel from the legal affairs department, and one researcher specialising in medical education research worked together as a team. They met regularly to discuss and solve discrepancies in the classification of cases according to their causes, specifically due to clinical reasoning errors.

For the categorisation of medical dispute cases associated with clinical reasoning errors, the team initially identified those with causes that were indirectly related to medical practice among all dispute cases. For example, these cases include equipment failure, lack of physician-patient communication or misunderstanding caused by inappropriate use of language, poor documentation of medical records, improper expectations of health outcomes by patients, complaints of healthcare cost, and attitude complaints of healthcare professionals. These were cases considered irrelevant to errors in clinical reasoning. Then, the team followed the definition of clinical reasoning [21] to determine whether the remaining cases were attributable to clinical reasoning errors. The cases due to clinical reasoning errors were further classified into skill- or knowledge-related categories. Misinterpretation of information, difficulties with prioritising clinical problems, or improper interventions (such as due to ignorance regarding disease-processes) were considered knowledge-related clinical reasoning errors. Skill-related clinical reasoning errors include faulty judgement regarding technical competence during operations or examinations such as resection of the wrong area, wrong-site surgery, or examinations. The frequency and characteristics of medical dispute cases were recorded and summarised using Microsoft Excel 2010 (Microsoft, Redmond, WA, USA). This study was carried out in accordance with relevant ethical guidelines and regulations and approved by the Chang Gung Medical Foundation Institutional Review Board on 6 June 2017, (Ref.: 201700740B0).

## 3. Results

A total of 78 medical dispute cases from the local hospitals during the 5-year period in Taiwan were identified, of which 45 (57.7%) were determined to be related to clinical reasoning errors after a thorough discussion. Among the 45 cases associated with clinical reasoning errors, 82.2% (37) and 22.2% (10) were knowledge- and skill-related errors, respectively (Figure 1). There are two clinical reasoning cases attributed to both aspects of lack of knowledge and technical competence. The proportion of clinical reasoning errors accounted for 57.7% of the medical dispute cases, indicating that more than half of these cases were due to clinical reasoning errors caused by different healthcare professionals.

In Table 1, different healthcare professionals were identified and associated with medical dispute cases. The attending physicians involved in 40 (51.3%) cases were the most frequently identified healthcare professionals, followed by the residents with 16 (20.5%) cases. In the clinical reasoning error related dispute cases, the attending physicians were also the leading healthcare professionals with the most dispute cases, with 25, (55.6%), followed by the residents (14, 31.1%), registered nurses (11, 24.4%), and technicians (2, 4.4%).

An analysis of the cause and effect of each case was determined by investigating the content of the disputes. A causality between medical disputes and medical practice was established in 25 out of 45 dispute cases (55.6%) related to clinical reasoning errors. There was no causal relationship between medical disputes and medical practice in the remaining 20 cases. Among all the 78 medical dispute cases, there were 19 cases (24.4%) that resulted in litigation, and 12 of which (63.2%) were related to clinical reasoning errors (Table 2).

In Table 3, the healthcare professionals related to the most medical dispute cases were obstetrician-gynecologists (10/90, 11.1%), surgeons (8/90, 8.9%), and emergency physicians (7/90, 7.8%). The top three occurrences of clinical reasoning error were also found in these three specialists, namely 7.8% (4/51) for obstetrician-gynecologists, 9.8% (5/51) for surgeons and 5.9% (3/51) for emergency physicians.

The number of clinical reasoning related medical disputes varied depending on the physician’s seniority. The attending physicians with a seniority of less than 5 years were found to have the highest number of healthcare professionals associated with medical dispute cases. In contrast, the highest seniority of over 25 years in the physician group and year 6 in the resident group are both shown with a zero number of medical dispute cases (Table 4).

The number of cases found in each hospital branch and the corresponding hospital size, denoted by the number of beds, were summarised in Table 5. There was a significant positive association between the number of dispute cases and the number of beds during the sampling period (r = 0.984, *p* < 0.05).

## 4. Discussion

There were 78 medical dispute cases internally assessed and retrieved from the Legal Affairs Department of this Taiwanese local hospital between 2011 and 2015. Although the percentage of medical disputes found in each year from 2011 to 2015 indicates a gradual decrease in medical dispute cases, 45 of which were dispute cases attributed to clinical reasoning (57.7%). In the analysis of 45 cases identified as related to clinical reasoning, a causal relationship between clinical reasoning errors and unintended patient outcomes was found in 25 of the cases. This means that out of all cases filed in the hospital, about 32% (25/78) of medical dispute cases are likely to be prevented by improving the clinical reasoning skills of the healthcare professionals through educational training, which in turn will avoid unintended adverse medical outcomes [26].

Healthcare professionals with various seniorities or healthcare disciplines might adopt different relevant clinical reasoning strategies [27]. Therefore, it is essential to consider factors in each case involving different healthcare professionals and explore the task categories and standards executed by every healthcare professional when analysing the cause of unexpected adverse events as a result of clinical reasoning errors [28,29]. In this study, among the 78 cases of medical disputes, obstetrics and gynecology, surgery, and emergency departments were the specialties with the highest frequency of medical disputes. In these specialties where medical disputes are more likely to occur, the proportion of clinical reasoning error-related cases is also relatively high. The common feature of these three specialties is the involvement of invasive treatment or examination during medical practice. The results reflect the content of the risk classification department of general physician medical liability insurance, which states that these specialties are characterised by high liability and high risk [30,31]. In particular, emergency medicine, which provides care for a wide range of undifferentiated patients, is likely to experience a high prevalence of dispute cases [32]. Therefore, specialties with a high incidence of medical errors are also vulnerable to the likelihood of errors in the clinical reasoning process [33]. Complex clinical settings with uncertainties expose physicians to the common pitfalls of cognitive failures causing clinical reasoning errors [34].

Despite the fact that the nature of certain specialties may be a key contributor to the incidence of medical disputes, the volume of hospital activities is also a factor to be taken into account. In contrast to a litigation study in Italy, where it shows that smaller hospitals were often associated with a greater number of claims [35], two local studies [6,30] in Taiwan have indicated that larger hospitals, such as medical centres and district hospitals, have a higher incidence of claims. The results from the Taiwanese studies support our finding that the number of dispute cases is correlated with hospital size, which is denoted by the number of beds. The larger hospital settings of medical centres and district hospitals in our study suggest the potential cause of medical dispute cases and, hence, the likelihood of reasoning errors.

In the analysis of seniority, the highest seniority of the attending physician group or the resident group shows the absence of the total number of unexpected adverse cases caused by clinical reasoning errors. Studies have shown that residents or trainees are more likely to be involved in medical lawsuits without proper supervision by their seniors [36,37]. Considering the two major teaching hospital branches included for data collection in our study, it is reasonable to observe the relatively high risk of medical reasoning errors associated with the individuals of less seniority in our result. The decline in the number of cases may be related to years of accumulating experience and expertise in learning and developing clinical reasoning skills [21,38]. With extensive experience and years of accumulated knowledge, expert physicians can categorise and process relevant information about disease comparatively faster than their junior colleagues [39]. This is congruent with the study on chest radiograph interpretation by Morra et al., where the authors demonstrated that expert physicians may provide better reasoning skills than their resident counterparts, resulting in potential positive patient outcomes [40]. The total number of cases shows little difference between residents in different seniority ranks, but this is because the number of residents fluctuates each year and therefore the proportion of the total residents cannot be calculated. Nursing staff are the second largest group of healthcare professionals associated with clinical reasoning errors. This could be explained by the fact that the number of nursing staff in hospitals is normally the highest among all categories of medical staff. The length of time spent with patients was also comparatively longer for nurses than for other healthcare professionals. It increases the likelihood of nurses being at risk of being involved in dispute cases when they deal with more patient interactions. In addition, two studies have indicated that nurses were significantly more likely to report the incidence of errors than physicians [41,42].

A limitation of this study is that not every clinical reasoning error falls neatly into one or the other of the binary classifications of skill- or knowledge-related categories. This is subject to the individual perceptions of the reasoning models of each researcher during reviews of the dispute cases. Their judgement during analysis might also be affected, especially when there was a lack of detail regarding the documentation of the dispute cases. However, these biases were substantially mitigated by efforts to ensure every researcher followed the same conceptual approach to classification through rigorous discussion.

## 5. Implications

The practice of the healthcare profession, which is an indispensable human endeavor, leaves healthcare professionals open to fallibility. According to the statistical records of the High Court of Taiwan, there were about 259–288 civil suits and about 60–95 criminal proceedings resulting from medical litigation cases each year between 2011 and 2015, giving rise to a total of about 326–354 cases being filed each year [43]. Consequently, the database from the hospitals in this study accounts for about one-twentieth of all medical dispute cases in Taiwan. The consequences of medical dispute cases can cause stress for healthcare providers and compromise patient safety and healthcare quality.

The following suggested measures should be undertaken to prevent medico-legal issues: update hospital management and regulations, establish a harmonious doctor-patient relationship, strengthen standard procedures of healthcare practice, keep detailed and complete medical records, use modern scientific and technological equipment, set up training courses (e.g., communication), and formulate medical dispute resolution mechanisms [44,45]. While the Objective Structured Clinical Examinations (OSCE) have been widely used for the teaching and assessment of clinical skills in healthcare trainees, one can design and develop OSCE clinical cases incorporating legal or ethical aspects based on real scenarios from these medical dispute cases.

Through OSCE-simulated training in medical ethics, it is proposed to be effective for students who might benefit from practical experience and immediate evaluation feedback in terms of understanding and learning medico-legal issues [46,47].

Having identified that clinical reasoning errors contributed a substantial fraction to the overall medical dispute cases, strategies to promote reasoning or diagnostic skills are essential. Studies have illustrated how knowledge acquisition, self-reflection, and check lists may mitigate the incidence of reasoning errors [26,48]. In addition to conventional classroom-based curricula for fostering clinical reasoning skills in healthcare trainees, such as case studies in combination with simulation, concept mapping, and debriefing [26,49], an emphasis should also be placed on workplace learning. Clinical reasoning is not only a construct of information gathering, processing, integration, and hypothesis generation leading to management plans but also a complex process in real clinical practice where cognitive activity, situational context, and social interactions with other healthcare professionals need to be considered [50,51]. One may devise approaches to examine the key components of reflection, heuristics, and metacognition in clinical reasoning by applying strategies that focus on exploration and immersion of healthcare practice within a socio-cultural context, such as video-reflexive ethnography and think-aloud [52].

## 6. Conclusions

An investigation and analysis of medical dispute cases reveal the frequency and nature of medical errors in Taiwanese hospitals. The identification of greater than half of the medical dispute cases attributable to clinical reasoning errors suggests that enhancement of clinical reasoning skills has become a major priority for all healthcare professionals. Increasing awareness of pitfalls and targeting improvement in clinical reasoning skills will fulfil the goals of reducing harm, making reliable diagnoses, and providing quality patient care.

## Figures and Tables

**Figure 1 healthcare-10-02224-f001:**
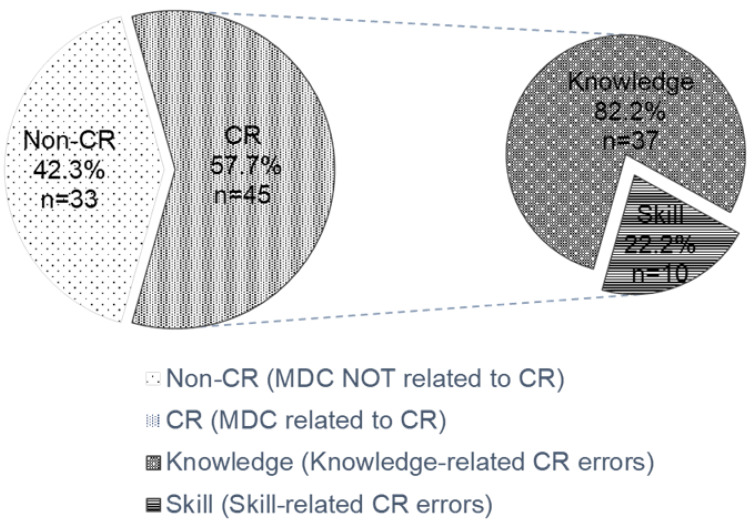
Medical dispute cases associated with clinical reasoning errors.

**Table 1 healthcare-10-02224-t001:** Characteristics of medical dispute cases from four local Taiwanese hospitals between 2011 and 2015.

	Dispute Cases		Clinical Reasoning Related Cases		
Year	No. of Cases	Proportion (%)	No. of Cases	Proportion (%)	Proportion of Clinical Reasoning Related Cases by Year (%)
2011	16	20.5	5	11.1	31.3
2012	19	24.4	11	24.4	57.9
2013	19	24.4	11	24.4	57.9
2014	16	20.5	11	24.4	68.8
2015	8	10.3	7	15.6	87.5
Total	**78**	100.0	**45**	100.0	57.7
	**Dispute Cases**		**Clinical Reasoning Related Cases**		
**Healthcare Professionals**	**No. of Cases ***	**Proportion (%)**	**No. of Cases ***	**Proportion (%)**	**Proportion of Clinical Reasoning Related Cases by Profession (%)**
Attending Physicians	40	51.3	25	55.6	62.5
Residents	16	20.5	14	31.1	87.5
Registered Nurses	13	16.7	11	24.4	84.6
Technicians	3	11.5	2	4.4	66.7

* Some dispute cases may involve more than one category of healthcare professionals.

**Table 2 healthcare-10-02224-t002:** Medical dispute cases that involved litigation between 2011 and 2015 and their causality with medical practice.

		Dispute Cases		Clinical Reasoning Related Cases	
Causality *	Yes	37	47.4%	25	55.6%
	No	41	52.6%	20	44.4%
Total		78		45	
Litigation	Yes	19	24.4%	12	26.7%
	No	59	75.6%	33	73.3%
Total		78		45	

* A causal relationship between medical dispute cases and medical practice.

**Table 3 healthcare-10-02224-t003:** Proportion of different specialties and healthcare professionals associated with medical dispute cases.

Specialties and Healthcare Professions	Number of Healthcare Professionals	Proportion	Healthcare Professionals Associated with Clinical Reasoning Dispute Cases	Proportion
Obstetrics and gynecology	10	11.1%	4	7.8%
General surgery	8	8.9%	5	9.8%
System-related (Instrument)	9	10.0%	1	2.0%
Emergency medicine	7	7.8%	3	5.9%
Nursing	5	5.6%	5	9.8%
Orthopedics	5	5.6%	5	9.8%
Plastic surgery	4	4.4%	2	3.9%
Internal medicine	4	4.4%	3	5.9%
Gastroenterology	3	3.3%	3	5.9%
Neurosurgery	3	3.3%	1	2.0%
Pulmonology	3	3.3%	1	2.0%
Anesthesiology	3	3.3%	1	2.0%
Dentistry	3	3.3%	2	3.9%
Neurology	3	3.3%	3	5.9%
Medical imaging	2	2.2%	1	2.0%
Traumatology	2	2.2%	1	2.0%
Psychiatry	1	1.1%	0	0.0%
Otorhinolaryngology	2	2.2%	1	2.0%
Radiology	2	2.2%	2	3.9%
Metabolism and endocrinology	1	1.1%	1	2.0%
Neonatology	1	1.1%	1	2.0%
Acupuncture	1	1.1%	0	0.0%
Cardiology	1	1.1%	1	2.0%
Cardiac surgery	1	1.1%	1	2.0%
Colorectal surgery	1	1.1%	1	2.0%
Health management center	1	1.1%	0	0.0%
Ophthalmology	1	1.1%	1	2.0%
Physical medicine and rehabilitation	1	1.1%	0	0.0%
Infectious disease	1	1.1%	0	0.0%
Pharmacy	1	1.1%	1	2.0%
Total	90	100%	51	100%

**Table 4 healthcare-10-02224-t004:** Number of physicians and residents associated with clinical reasoning error cases with respect to their different seniorities.

Job Position	Seniority (Years)	Number
Physicians	Over 25	0
	21–25	5
	16–20	3
	11–15	6
	5–10	5
	Less than 5	11
Residents	R6	0
	R5	2
	R4	5
	R3	3
	R2	5
	R1	2

**Table 5 healthcare-10-02224-t005:** Correlation between number of dispute cases and hospital size (number of beds) found across four hospital branches.

Hospital Branches	Number of Cases	Number of Beds
**North centre**	43	4176
**North district**	1	1089
**South centre**	22	2680
**South district**	12	1384

Pearson Correlation Coefficient = 0.984, *p* = 0.016 (*n* = 4); significance was defined as *p* < 0.05.

## Data Availability

All data generated or analysed during this study are included in this published article.

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
