# Peer review of "Medical Dispute Cases Caused by Errors in Clinical Reasoning: An Investigation and Analysis"

_healthcare, 2022, doi:10.3390/healthcare10112224_

Round 1

Reviewer 1 Report (Previous Reviewer 1)

I read the new version of the manuscript with interest. I congratulate you on the work done.

Kind Regards

Author Response

We appreciate the reviewer's positive comment and support.

Reviewer 2 Report (Previous Reviewer 2)

I suggest the authors to include in the summary the result of the research that shows that larger hospitals have higher incidence of claims compared to smaller hospitals.

Author Response

Thank you, we have now added the following statement to Abstract as follows (Line 22): In our study, the larger hospitals have significantly higher incidence of MDC compared to the smaller hospitals (Pearson’s Correlation=0.984, p=0.016).

This manuscript is a resubmission of an earlier submission. The following is a list of the peer review reports and author responses from that submission.

Round 1

Reviewer 1 Report

Thank you for submitting the manuscript. I have read your paper with great attention and interest. Your topic is important and little investigated. However, there are some points that need to be clarified.In the methods you cite a department of a hospital in Taiwan that should also serve as a collection point for the cases of the other hospitals. It seems to me an unclear point. First of all, the data you present are aggregated and not mentioning the name of the hospitals involved seems to me a serious mistake. Let me explain. There is literature, and I will provide you with references, which explains how the volume of hospital activity also affects the number of disputes. I suggest the references:doi: 10.3390/healthcare10071328. doi: 10.1001/jamasurg.2017.2979. doi: 10.3390/healthcare9081012.

I therefore ask you to specify the names of the hospitals, the characteristics (university, private, etc.), the number of beds. This way you can analyze whether there is a correlation between the complaints and the volume of activity.

I also ask you to include in the paper the competent Ethics Committee and the study approval number.

I hope my comments are useful to you.

Kind Regards

Reviewer 2 Report

I suggest the authors to describe in more detail the methodology of classifying cases into those related to CR versus those non CR.
Also emphasize in the discussion the potential errors in reasoning that results from the possible subjectivity of the three researchers.

Round 2

Reviewer 1 Report

Thank you for submitting the manuscript. I am only partially satisfied with the corrections you have made. I repeat that the volume of activity of a center is closely correlated with the incidence of errors, as it affects organization, experience and skills. Please reformulate the paragraph, taking into account the references I suggest that you have ignored instead.

I hope my comments are useful to you.

Kind Regards

Author Response

Thank you. We have now taken into account the 3 references provided and made extra points in the discussion. Please see Line 196 and Line 205. Two new citations were also added to support the relative evidence.